# Preparation and Anti-Lung Cancer Activity Analysis of Guaiacyl-Type Dehydrogenation Polymer

**DOI:** 10.3390/molecules28083589

**Published:** 2023-04-20

**Authors:** Junyi Zhou, Yuanyuan Yue, Xin Wei, Yimin Xie

**Affiliations:** 1Research Institute of Pulp & Paper Engineering, Hubei University of Technology, Wuhan 430068, China; z2803867194@163.com (J.Z.);; 2Hubei Provincial Key Laboratory of Green Materials for Light Industry, Hubei University of Technology, Wuhan 430068, China

**Keywords:** lignin, dehydrogenation polymer, classified fraction, purified compound, anti-tumor activity, chemical structure

## Abstract

In this paper, guaiacyl dehydrogenated lignin polymer (G-DHP) was synthesized using coniferin as a substrate in the presence of β-glucosidase and laccase. Carbon-13 nuclear magnetic resonance (^13^C-NMR) determination revealed that the structure of G-DHP was relatively similar to that of ginkgo milled wood lignin (MWL), with both containing β-O-4, β-5, β-1, β-β, and 5-5 substructures. G-DHP fractions with different molecular weights were obtained by classification with different polar solvents. The bioactivity assay indicated that the ether-soluble fraction (DC_2_) showed the strongest inhibition of A549 lung cancer cells, with an IC_50_ of 181.46 ± 28.01 μg/mL. The DC_2_ fraction was further purified using medium-pressure liquid chromatography. Anti-cancer analysis revealed that the D_4_ and D_5_ compounds from DC_2_ had better anti-tumor activity, with IC_50_ values of 61.54 ± 17.10 μg/mL and 28.61 ± 8.52 μg/mL, respectively. Heating electrospray ionization tandem mass spectrometry (HESI-MS) results showed that both the D_4_ and D_5_ were β-5-linked dimers of coniferyl aldehyde, and the ^13^C-NMR and ^1^H-NMR analyses confirmed the structure of the D_5_. Together, these results indicate that the presence of an aldehyde group on the side chain of the phenylpropane unit of G-DHP enhances its anticancer activity.

## 1. Introduction

In 2020, there were approximately 19.3 million cancer patients worldwide, and 10 million people died of cancer. At present, lung cancer remains the most lethal cancer [1,2]. Chemotherapy is the most common therapeutic approach used to fight lung cancer. However, it has several limitations, including low bioavailability, poor water solubility, a low therapeutic index, high dose requirements, and poor targeting [3]. In addition, chemotherapy has severe adverse effects on the human body. Therefore, there is a critical need to identify natural drugs that can be applied in the treatment of lung cancer.

Lu et al. [4] found that alkali-treated lignin inhibited enzymatic and non-enzymatic lipid peroxidation, hindered the activity of glucose-6-phosphate dehydrogenase (G6PD), which is an enzyme implicated in the generation of superoxide anion radicals, and reduced the growth of HeLa S3 cervical cancer cells. Zhang et al. [5] showed that lignin macromolecules can promote cancer cell apoptosis and inhibit the expression of the transcription factor nuclear factor-k gene binding (NF-κB) in cancer cells. Lábaj et al. [6] separated lignin from the waste liquor of kraft pulping and determined the anti-tumor activity of lignin in mouse lymphocytes. As compared with the control group, the DNA strand damage induced by H_2_O_2_ was significantly reduced, i.e., the resistance to H_2_O_2_-induced oxidative stress was enhanced. Geies et al. [7] used the in situ sodium hydroxide-sodium bisulphate method to separate lignin from three different agricultural and industrial wastes, i.e., sweet sorghum, rice straw, and bagasse. The results indicated that all lignin samples had high cytotoxicity potential on A549 lung cancer cells (IC_50_ = 12~17 µg/mL).

Lignin also has good biocompatibility. Sakagami et al. [8,9] found that after oral administration of labelled natural lignin and synthetic lignin to mice, lignin was absorbed through the digestive system and excreted through the urine and feces, and did not remain in the body. Ugartondo et al. [10] found that bagasse lignin had a high antioxidant capacity within a certain concentration range and was harmless to normal human cells. Barapatre et al. [11] used pressurized solvent extraction (PSE) and continuous solvent extraction (SSE) to extract different lignin components from acacia wood. The authors found that these components had high cytotoxicity potential on MCF-7 human breast cancer cells (IC_50_: 2~15 µg/mL), but only had significant inhibitory effects on normal primary human hepatic stellate cells (HHSteCs) when the concentration was greater than 100 µg/mL. A low-molecular-weight lignin sample was found to have stronger anti-human immunodeficiency virus type 1 (HIV-1) performance than high-molecular-weight lignin [12]. Further, synthetic lignin dimer-like compounds with a β-5′ structure displayed stronger inhibitory activities [12]. Together, findings in the literature demonstrate that lignin has anti-tumor, antibacterial, antiviral, antioxidant, and other physiological activities [8,9,10,11,12,13,14,15]. However, due to its complex structure, the source of lignin’s anticancer activity has not yet been fully identified. Studies have examined the effects of lignin dehydrogenated polymer (DHP) on breast cancer cells (MCF7), normal fetal lung fibroblasts (MRC5) [16], and cervical cancer HeLa cells [17], and DHP has been found to contain β-5. Dimer and trimer DHP with a β-5 structure can inhibit the growth of cervical cancer cells; the β-O-4 ether bond does not participate in this anticancer activity, while the carboxyl group helps to increase the inhibition effect of DHP on cervical cancer cells [17]. Further, low-molecular-weight DHP can inhibit the growth of MRC5 human embryonic lung cells [16]. Therefore, human malignant tumor cells are highly sensitive to DHP, which further demonstrates the potential value of DHP as an anticancer drug. However, there remains a paucity of studies on the anti-lung cancer activities of DHP, especially low-molecular-weight DHP with a guaiacyl substructure.

In this study, guaiacyl-type DHP was synthesized and extracted with different solvents. The phenolic compounds with good anti-lung cancer effects were separated by medium-pressure chromatography. Their anti-tumor activities against A549 lung cancer cells were analyzed. Furthermore, their chemical structures were analyzed by mass spectrometry (MS), carbon-13 nuclear magnetic resonance (^13^C-NMR), and proton nuclear magnetic resonance (^1^H-NMR), and the structure–activity relationship of lignin’s biological activity of lignin was investigated. These findings offer novel ideas for exploring the commercial application of the DHP.

## 2. Results and Discussion

### 2.1. ^13^C-NMR Spectral Analysis of DHP

The ^13^C-NMR spectrum of G-DHP is shown in Figure 1. A weak γ-CHO signal of cinnamaldehyde can be observed at 194.2 ppm (No. 1), indicating that a small amount of oxidation of G-DHP occurs during the polymerization process [18]. At 172.1 ppm (No. 2), the signal of ferulic acid at the γ-position of ferulic acid is observed, indicating that the hydroxymethyl group at the γ-position is oxidized to form cinnamic acid [19,20]. The signals at 169.5 ppm (No. 3), 149.9 ppm to 132.3 ppm (No. 4 to No. 9), 118.7 ppm (No. 15), and 115.0 ppm (No. 16) are of carbon atoms on the guaiacyl aromatic ring. The signals at 85.3 ppm (No. 19), 53.7 ppm (No. 28), and 67.3 ppm (No. 24) are of Cα, Cβ, and Cγ in the β-5 structure, respectively [21,22,23]. A peak at 87.3 ppm (No. 18) mainly reflects Cβ in the β-O-4 structure and Cα in the β-β structure. The signals at 73.6 ppm to 70.2 ppm (No. 21 to No.23) and 61.7 ppm (No. 26) are of Cα and Cγ in the β-O-4 structure [24], while the signal at 63.0 ppm (No. 25) is of Cα in the β-1 structure. The signal at 45.2 ppm (No. 29) is of C-β in the β-β structure [22]. These results are in general agreement with those of previous studies [17,25]. Together, these findings suggest that the synthesized G-DHP contained mainly β-O-4 and β-5 structures, but also β-β, β-1 and 5-5 structures and a small amount of cinnamic aldehyde and ferulic acid.

### 2.2. Analysis of Molecular Weights of the DHP Fractions

The molecular weights of each classified fraction of G-DHP are shown in Table 1. It can be found that the molecular weights of the classified fractions of the G-DHP increased with the increasing polarity and solubility of the solvent. The molecular weights of the G-DHP ranged from 250 Da to 3500 Da.

### 2.3. Analysis of the Total Phenolic Content (TPC) of DHP-Classified Fractions

The phenolic hydroxyl groups in phenolic compounds under alkaline conditions will change the Folin-Phenol reagent from yellow to blue. By measuring the absorbance of the standard sample at different concentrations, a standard curve was established, and the TPC of the experimental samples relative to the standard was calculated by substituting the absorbance of each sample into the standard curve equation.

The relationship between the concentration of gallic acid and absorbance at 760 nm is shown in Figure 2. The TPC is expressed as mg of gallic acid per g of sample. The concentration of gallic acid was linearly related to the absorbance, and the equation of the standard curve was as follows: y = 0.9581x + 0.07432, R^2^ = 0.98106. The absorbance of each classified component of the G-DHP was substituted into the equation to obtain the concentration. The TPC of each classified component relative to gallic acid is presented in Table 2. The TPC of each classified component decreased with increasing molecular weight.

### 2.4. Analysis of the Anti-Tumor Activities of the G-DHP Classified Fractions

The effects of the G-DHP classified fractions and coniferin, at a concentration of 800 μg/mL, on the growth of A549 lung cancer cells are shown in Figure 3. It can be found that the lignin precursor coniferin did not have any obvious inhibitory effect on lung cancer cells. Meanwhile, the cell growth status of the control group, using the solvent only, was also normal, indicating that the solvent had no effect on the growth of A549 lung cancer cells. The inhibitory effects of the classified fractions of the G-DHP on A549 lung cancer cells were significant, especially for the ether-soluble fraction DC_2_. Specifically, the cells became round and contracted and no longer adhered to the wall, and some even broke into small fragments. Furthermore, many apoptotic cells died.

The relationships between the concentration of the classified fractions and the inhibition rate on A549 lung cancer cells are shown in Figure 4. The IC_50_ values are shown in Table 3. It can be found that all classified fractions inhibited A549 lung cancer cells. Furthermore, the relationships between the concentration and the inhibition rate were approximately logarithmic. The ether-soluble fraction (DC_2_) showed the strongest inhibitory effect on A549 lung cancer cells, with an IC_50_ of 181.46 ± 28.01 μg/mL. According to the standards of the National Cancer Institute. IC_50_ ≤ 20 μg/mL is highly cytotoxic; ranging from 21 to 200 μg/mL is moderately cytotoxic; ranging from 201 to 500 μg/mL is weakly cytotoxic; and an IC_50_ > 501 μg/mL is noncytotoxic [26,27,28].

### 2.5. Analysis of the Anti-Tumor Activity of the Purified Substances of G-DHP Ether-Soluble Fraction

The effects of the purified substances from the DC_2_ fraction, at a concentration of 800 μg/mL, the growth state of A549 lung cancer cells are shown in Figure 5. It can be found that the A549 lung cancer cells were apoptotic in the experimental groups with the addition of each purified substance. The cells became smaller and rounder, the space between the cells became larger, and the cells no longer adhered to the wall. Some of the apoptotic cells broke off into small fragments. The purified substances exhibited different degrees of inhibitory effects on A549 lung cancer cells, with compound D_5_ exhibiting the best inhibitory effect.

The relationships between the different concentrations of purified compounds from the DC_2_ fraction and the inhibition rate on A549 lung cancer cells are shown in Figure 6. The IC_50_ values are shown in Table 4. The results revealed that all purified compounds had inhibitory effects on A549 lung cancer cells, while the concentrations and inhibition rates exhibited roughly logarithmic relationships. Obvious inhibitory effects of D_4_ and D_5_ on A549 lung cancer cells were observed, with IC_50_ values of 61.54 ± 17.10 μg/mL and 28.61 ± 8.52 μg/mL, respectively. As compared with the original DC_2_ fraction, the purified substances showed significantly enhanced inhibitory effects on A549 lung cancer cells, indicating that the purification by medium-pressure liquid preparative chromatography was successful for the isolation of the bioactive substances. In our previous research, some compounds isolated from DHP had a certain inhibitory effect on the HeLa cervical cancer cells. The inhibitory property of a compound, i.e., 4-[3-Hydroxymethyl-5-(3-hydroxy-propenyl)-7-methoxy-2,3-dihydro-benzofuran-2-yl]-2-methoxy-phenol from ether-soluble fractions, were relatively strong, with IC_50_ values of 15.1 ± 2.3 μg/mL. This compound was a typical (β-5) G-type dimer, and very similar to compounds D4 and D5 in structure in the present work. This also meant that the D4 and D5 compounds are relatively effective in anti-cancer treatments [17]. We have conducted a large number of experiments on the metabolic activity of hepatocytes cultured in vitro in the lignin and lignin-carbohydrate complexes. The growth of normal hepatocytes cells was enhanced by the addition of lignin and linin-carbohydrate complexes, and the inhibition effect could be ignored [29,30].

### 2.6. HESI-MS Mass Spectrometry Analysis of the Purified Substances

Electrospray ionization mass spectrometry (ESI-MS) is appropriate for investigating the primary and fragment structure of biopolymers. Several previous studies on lignin that used ESI-MS have shown the utility of this analytical method for estimating the average molecular mass of lignin macromolecules, distinguishing their biological origins, and exploring the reactions occurring in solutions [31,32]. Given the high contents of hydroxyl and carboxyl groups in lignin, mass spectrometric information in the negative ion mode (M − H^+^) will produce better signals than in the positive ion mode (M + H^+^). Therefore, ESI under negative ionization conditions is the most widely used method for determination of lignin structure [33]. Coniferin was enzymatically hydrolyzed to coniferyl alcohol before polymerization to DHP. The molecular weight of the monomer coniferyl alcohol was about 180 g/mol. In addition, functional groups such as carboxyl and aldehyde groups may be generated under the catalytic action of laccase, thus, the molecular weight distributions of DHP with different degrees of polymerization were as follows: monomer (130–190 g/mol), dimer (190–380 g/mol), trimer (380–570 g/mol) [34,35,36].

The mass spectral information of the purified substance D_5_ from the DC_2_ fraction of G-DHP is shown in Figure 7. The signal at *m*/*z* 353.103 is of the molecule D_5_. The molecular structure formula of D_5_ was calculated to be C_20_H_18_O_6_ based on the other fragments The most abundant ion fragment is at *m*/*z* 149.060, which is the ion peak of 2-methoxy-4-vinyl coniferyl alcohol. This ionic peak originates from the cleavage of the coumarin structure, indicating that the Cγ on the side chain of the phenyl propane monomer in guaiacyl lignin is unstable as compared to other positions. The signal at *m*/*z* 137.0 is of the fragment peak of 2-methoxy-4-methylphenol formed by the elimination of Cβ on the side chain of the fragment at *m*/*z* 149.060. This is consistent with previous findings, where *m*/*z* 177.0 was found to be the fragment peak of 2-methoxy-4-methylphenol [37]. The ionic peak of the coniferyl alcohol monomer at *m*/*z* 177.0 is also consistent with previous findings [38]. The peak at *m*/*z* 353.1 is of the β-5-linked dimer of coniferyl aldehyde (β-5, γ-CHO, γ’-CHO), while the peak at *m*/*z* 325.1 differs from *m*/*z* 353.1 by 28, indicating that *m*/*z* 325.1 is a fragment peak formed by the elimination of the aldehyde group on Cγ of the molecular ion at *m*/*z* 353.1. These results indicate that the oxidation of coniferin catalyzed by laccase and β-glucosidase led to the formation of the β-5 structure of the coniferyl aldehyde dimer. Similar findings have been reported in previous studies that have synthesized DHP [39].

The mass spectrum of the purified substance D_4_ from the DC_2_ fraction of G-DHP is shown in Figure 8. Similarly to D_5_, the molecular ion of D_4_ is also found at *m*/*z* 353.103. Considering the other fragments, the molecular structure formula of D_4_ should also be C_20_H_18_O_6_. All of the fragment ion peaks in D_5_ can be found in the spectrum of compound D_4_, while D_4_ has additional peaks at *m*/*z* 129.018, *m*/*z* 167.034, *m*/*z* 187.061, *m*/*z* 203.056, and *m*/*z* 301.072, indicating that D_4_ contains some impurities. These results indicate that D_4_ and D_5_ are generally the same substance, but D_4_ contains more impurities. This analysis is also confirmed by the slightly lower inhibitory effect of the compound D_4_ on A549 cells, as compared to the compound D_5_.

The mass spectrum of the purified substance D_4_ from the DC_2_ fraction of G-DHP is shown in Figure 8. Similar to that of D_5_, the molecular ion of D_4_ is also found at *m*/*z* 353.103. Considering the other fragments, the molecular structure formula of D_4_ should also be C_20_H_18_O_6_, and all of the fragment ion peaks in D_5_ can be found in the spectrum of compound D_4_, while D_4_ has additional peaks at *m*/*z* 129.018, *m*/*z* 167.034, *m*/*z* 187.061, *m*/*z* 203.056, and *m*/*z* 301.072, indicating that D4 contains some impurities. The results indicate that D_4_ and D_5_ are mainly the same substance, but D_4_ contains more impurities. This is also confirmed by the slightly less inhibitory effect of D_4_ on A549 cells than D_5_.

### 2.7. ^13^C-NMR and ^1^H-NMR Analysis of the Purified Compound

The ^13^C-NMR spectrum of the purified compound D_5_ from the DC_2_ fraction of G-DHP is shown in Figure 9. The major resonance signals are assigned as follows: δ 193.81 (Cγ), δ 192.81 (Cγ′), δ 153.60 (Cα′), δ 148.13 (C4′), δ 147.32 (C3), δ 146.75 (C4), δ 145.12 (C3′), δ 131.61 (C1), δ 129.72 (C1′), δ 129.43 (C5′), δ 127.03 (Cβ′), δ 120.94 (C6), δ 120.67 (C6′), δ 115.57 (C2′), δ 115.39 (C5), δ 111.51 (C2), δ 88.30 (Cα), δ 60.32 (Cβ), δ 56.52 (C7′), δ 56.16 (C7), δ 40.05 (DMSO).

The ^1^H-NMR spectrum of the purified compound D_5_ from the DC_2_ fraction of G-DHP is shown in Figure 10. The major resonance signals are assigned as follows: δ 9.88 (7), δ 9.70 (1), δ 8.24 (12), δ 7.61 (3), δ 7.59 (6), δ 7.17 (4), δ 6.98 (10), δ 6.96 (14), δ 6.91 (2), δ 6.86 (13), δ 5.63 (9), δ 4.28 (8), δ 3.87 (5), δ 3.81 (11), δ 2.49 (DMSO). Thus, considering the mass spectrum, ^13^C-NMR spectrum, and ^1^H-NMR spectrum analyses, it can be concluded that compound D_5_ is a β-5-linked coniferyl aldehyde dimer.

### 2.8. Discussion of the Structure–Effect Relationships of the Purified Substances

The HESI-MS, ^13^C-NMR, and ^1^H-NMR results indicated that compound D_5_ is a coniferyl aldehyde dimer with a β-5 linkage. The compounds D_4_ and D_5_ are the same substance, although the compound D_4_ contains more impurities, leading to a reduction in its anti-tumor activity. Zemek et al. [40] studied the biological activity of lignin model substances and found that those containing Cα=Cβ on the side chain of the benzene ring and with methyl on Cγ were the most effective. In addition, increases in hydroxyl, carbonyl, and carboxyl groups on the side chain reduced the biological activity of the compound. Xie et al. [17,41] synthesized DHP from isoeugenol and found that the presence of an aldehyde group on the side chain of the phenylpropane structure weakened the antioxidant and antibacterial properties of DHP. However, the results of the present study indicated that compound D_5_ contains aldehyde groups on the side chain, yet it had strong anti-tumor activity. Gładkowski et al. [42] used aromatic aldehydes as raw materials to synthesize 20 aromatic aldehydes. The authors found that some of the obtained compounds had significant cytotoxic effects on two cancer cells: human leukemia cells (Jurka cells) and canine osteosarcoma cells (D_17_ cells). These results suggest that aldehydes have potential anti-tumor applications.

Previous studies have shown that β-5-linked oligomeric lignins have good inhibitory effects on cervical cancer HeLa cells, with IC_50_ values as low as 15.1 μg/mL [17], while the present study found that β-5-linked oligomeric lignin had an inhibitory effect on A549 lung cancer cells, indicating that β-5 structured oligomeric lignins have broad anticancer activity.

## 3. Experiment

### 3.1. Materials

With reference to the method of Xie et al. [22], coniferin was synthesized from vanillin. The synthetic route is shown in Figure 11. β-Glucosidase was purchased from Sigma Co., Ltd. (Sigma, St. Louis, MO, USA) and laccase (No. 51003) was purchased from Novazyme Co., Ltd. (Tianjin, China). Complete Medium (CM2-1) was composed of 89% Roswell Park Memorial Institute (RPMI) 1640 basal medium (Gibco, Grand Island, NY, USA), 10% fetal bovine serum (FBS) (Tianhang, China), and 1% penicillin-streptomycin mixture (Gibco, USA). Other chemicals were of analytical grade and were purchased from Sinopharm Chemical Reagent Co., Ltd. (Shanghai, China). Human A549 lung cancer cells were purchased from Beijing Beina Biotechnology Co., Ltd. (BNCC, Henan Xinyang, China).

### 3.2. Methods

#### 3.2.1. Synthesis of DHP

The DHP synthesis mechanism is shown in Figure 12. In 150 mL of 0.2 mol/L sterile acetic acid/sodium acetate buffer solution with pH 4.83, 5 g of coniferin, 50 mg of β-glucosidase (6.4 U/mL), and 3 mL of laccase (648 U/mL) were added with the continuous blowing of sterile air filtered by activated carbon with an air pump. Then, the mixture was allowed to continue reacting in a water bath at 30 °C for 25 min. After the reaction was completed, 150 mL of distilled water was added and then heated to 70 °C to stop the reaction. The mixture was then centrifuged, the precipitate collected and washed with distilled water more than 5 times to remove the unreacted coniferin, and some enzymes freeze-dried to obtain crude DHP. The mixture was dissolved in dichloroethane/ethanol (*v*/*v* = 2:1), and the precipitate of the residual enzyme was removed by centrifugation. The solvent was then removed in vacuo. The DHP was obtained by vacuum drying with a yield of 38.36%.

#### 3.2.2. ^13^C-NMR Measurement of G-DHP

An 80 mg DHP sample was placed into φ 5 mm NMR tubes and dissolved with 0.6 mL of DMSO-d_6_. A 600-dd2 NMR spectrometer (Agilent Technologies, Santa Clara, CA, USA) was used to scan the solution at 150.90 MHz to obtain the corresponding ^13^C-NMR spectrum. The parameters of the instrument were as follows: test temperature: 25 °C; pulse delay: 2.0000 s; acquisition time: 0.6900 s; and number of scans: 3000.

#### 3.2.3. Classification of G-DHP

G-DHP was classified according to the polarity and solubility of different organic solvents, with reference to the method of Li et al. [43]. G-DHP was graded using n-hexane, ether, ethyl acetate, and dioxane. The classification process is shown in Figure 13. G-DHP was placed into a Soxhlet extractor together with n-hexane and heated at boiling point to reflux for 6 h. The extracted solution was distilled in vacuo to remove the solvent and then dried under a vacuum to obtain the n-hexane-soluble component (DC_1_). The remaining solid in the Soxhlet extractor was heated and refluxed with ether at 40 °C for 6 h. The extracted solution was distilled in vacuo to remove the solvent and then dried under a vacuum to obtain the ether-soluble component (DC_2_). The ethyl acetate-soluble component (DC_3_) and dioxane-soluble component (DC_4_) were obtained by repeating the above procedures with ethyl acetate and dioxane, respectively. Finally, the yields of the n-hexane-soluble component (DC_1_), ether-soluble component (DC_2_), ethyl acetate-soluble component (DC_3_), and dioxane-soluble component (DC_4_) were 6.8%, 11.7%, 43.6%, and 32.3%, respectively.

#### 3.2.4. Determination of Molecular Weight of the G-DHP Fractions

The relative molecular weights of individual fractions were determined by gel permeation chromatography (GPC). The molecular weight standard curves were prepared with EasiVial Polystyrene PS-M. Each DHP fraction (2 mg) was dissolved in 2 mL N,N-dimethylformamide (DMF) filtered through a 0.22 μm microporous membrane, and then injected into the GPC system (20AT, Shimadzu, Osaka, Japan) for molecular weight measurement. The instrument parameters were as follows: column: Stirrage HR 4E DMF (7.8 × 300 mm) with a guide column of Styrage DMF (4.6 × 30 mm); detector: RID-10A differential refraction detector; mobile phase: DMF (containing 0.1 mol/L LiCl); flow rate: 1.0 mL/min; column temperature: 50 °C; injection volume: 20 μL.

#### 3.2.5. Determination of Total Phenol Content (TPC)

The Folin–Ciocalteu colorimetric method [44] was used to determine the TPC of the various fractions. For the quantification of these extractives, a standard curve was established from the analysis of different concentrations of gallic acid varying from 125–1000 μg/mL. Then, 100 µL methanol extractive was added to test tubes with 500 µL Folin–Ciocalteu reagent. Six milliliters of distilled water was then added. After shaking for 1 min, 2 mL of 15% Na_2_CO_3_ solution was transferred to the test tubes within 0.5~8 min. Finally, distilled water was added to the tubes to make a total volume of 10 mL. After being kept in the dark at room temperature for 2 h, the test tubes were homogenized with ultrasonic agitation. The spectrophotometer (Shimadzu 2550, Nakagyo-ku, Kyoto, Japan) was set at a wavelength of 760 nm, and the absorbance reading was measured. The TPC of each sample was calculated according to the following formula:TPC [mg (gallic acid)/g(sample)] = GAE × FV × DF/SW

Legend: GAE is the concentration of gallic acid calculated from the standard curve (mg/mL); FV is the final volume of the sample extract (mL); DF is the dilution multiple; SW is the sample mass (g).

#### 3.2.6. Determination of the Anti-Tumor Activities of DHP Fractions and Purified Compounds

According to the method of Ahamed, Le et al. [45,46], the antitumor activities of the G-DHP fractions and purified compounds were determined using the 3-(4,5-Dimethylthiazol-2-yl)-2,5-diphenyltetrazolium bromide (MTT) method. First, samples were dissolved in acetone: CM2-1 medium (0.99:99.1 *v*/*v*, 0.1% polysorbate 80 as dispersant) to obtain a series of solutions with concentration in the range from 12.5 μg/mL to 800 μg/mL. Second, 5 × 10^3^ exponentially growing tumor cells were suspended in 100 μL of the CM2-1 medium and plated per experimental well on 96-well plates. Blank wells were also filled with 200 µL of complete medium. Third, the cells were cultured in a carbon dioxide cell incubator (37 °C, 5% CO_2_). After 16–48 h, 100-μL sample solutions with different concentrations (12.5~800 μg/mL) were transferred to every experimental well, which was different from the sample solvent added to the control wells. After incubation for 48 h, 20 μL MTT solution (5 mg/mL) was added to each well. Four hours later, the liquid in each well was discharged with a sterile syringe, and then 150 μL of dimethyl sulfoxide (DMSO) was added to each well. Finally, the plate was placed into a microplate reader after shaking at a low speed for 10 min. The absorbance of each well was measured at 490 nm. The inhibition rate of each sample was calculated using the following equation:IR = (A_CONTROL_ − A_EXPERIMENTAL_)/(A_CONTROL_ − A_BLANK_) × 100%;
CONTROL = medium without sample containing cells, solvent, MTT and DMSO;
BLANK = medium without cells and sample containing, solvent, MTT and DMSO.

Then, SPSS 26 software was used to calculate the semi-inhibition concentration (IC_50_) of each sample [47].

#### 3.2.7. Purification of the Ether-Soluble Fraction of DHP with Preparative Column Chromatography

The ether-soluble component DC_2_ of DHP was gradient eluted and repeatedly purified using a medium-pressure liquid preparation chromatograph (Buchi C-615, Zurich, Switzerland) equipped with an ultraviolet (UV) detector at a wavelength of 280 nm. The stationary phase was Sephadex LH-20 gel (100~200 mesh), and the mobile phase was a chloroform-methanol mixed eluent (the concentration gradient of methanol was in the range of 0% to 50%), with a flow rate of 2.5 mL/min.

First, the samples were dissolved in a 50% chloroform-methanol solution, and then the sample solution was injected into the dosing ring using a glass syringe. After eluting the separation column with 50% chloroform-methanol, the concentration of methanol was gradually increased, and finally, the column was completely washed with methanol. During the separation process, a UV 230 II ultraviolet-visible detector was applied. The ether-soluble component DC_2_ of DHP was purified to obtain compounds D_1_, D_2_, D_3_, D_4,_ and D_5_, with yields of 7.47%, 41.67%, 29.91%, 10.83%, and 10.12%, respectively.

#### 3.2.8. Mass Spectrometry Analysis of the Structure of the DHP Purified Compound

The mass spectrum information of D_5_ was obtained using electrostatic field orbital hydrazine mass spectrometry (Thermo Fisher Scientific, Waltham, MA, USA). The ion source was HESI. The determination conditions were as follows: dry gas: N_2_; spray voltage: 3200 V; temperature of the ion transmission capillary: 300 °C; auxiliary gas temperature: 280 °C; scanning mode: full MS/dd-MS2; scanning range *m*/*z*: 50–750.

#### 3.2.9. ^13^C-NMR and ^1^H-NMR Measurement of the Purified Substance D_5_

##### ^13^C-NMR Measurement of the Purified Substance D_5_

An 80 mg purified substance D_5_ sample was placed into φ 5 mm NMR tubes and dissolved with 0.6 mL of DMSO-d_6_. A 600-dd2 NMR spectrometer (Agilent Technologies, Santa Clara, CA, USA) was used to scan the solution at 150.90 MHz to obtain the corresponding ^13^C-NMR spectrum. The parameters of the instrument were as follows: test temperature: 25 °C; pulse delay: 2.0000 s; acquisition time: 0.6900 s; and number of scans: 3000.

##### ^1^H-NMR Measurement of the Purified Substance D_5_

A 15 mg volume of the purified substance D_5_ sample was placed into φ 5 mm NMR tubes and dissolved with 0.6 mL of DMSO-d_6_. A 600-dd2 NMR spectrometer (Agilent Technologies, Santa Clara, CA, USA) was used to scan the above sample at 600 MHz to obtain the corresponding ^1^H-NMR spectrum. The parameters of the instrument were as follows: test temperature: 25 °C; pulse delay: 5.0000 s; acquisition time: 1.4500 s; and number of scans: 256.

## 4. Conclusions

(1)^13^C-NMR determination revealed that the structure of G-DHP was relatively similar to that of the gymnosperm ginkgo MWL, which was dominated by β-O-4, β-5, β-1, β-β, and 5-5 substructures.(2)G-DHP was classified with different polar solvents to obtain a hexane-soluble fraction (DC_1_), ether-soluble fraction (DC_2_), ethyl acetate-soluble fraction (DC_3_), and dioxane-soluble fraction (DC_4_). The GPC results and total phenolic content measurements showed that the total phenolic content of fractions decreased with an increase in molecular weight. The bioactivity assay showed that the ether-soluble fraction (DC_2_) had the strongest inhibitory effect on lung cancer A549 with an IC_50_ of 181.46 ± 28.01 μg/mL.(3)Further purification of the DC_2_ fraction revealed that the obtained compounds D_4_ and D_5_ had significant anti-tumor activities, with IC_50_ values of 61.54 ± 17.10 μg/mL and 28.61 ± 8.52 μg/mL against lung cancer A549, respectively. HESI-MS measurements indicated that D_5_ was a coniferyl aldehyde dimer linked by β-5. The compounds D_4_ and D_5_ were similar substance, but D_4_ contained more impurities. ^13^C-NMR and ^1^H-NMR results also confirmed the chemical structure of D_5_.

In combination with the results of the anti-tumor assay, for the small molecule compound of guaiacyl type DHP, it was found that the presence of an aldehyde group on the side chain of the phenylpropane subunit can enhance the anti-cancer activity.

## Figures and Tables

**Figure 1 molecules-28-03589-f001:**
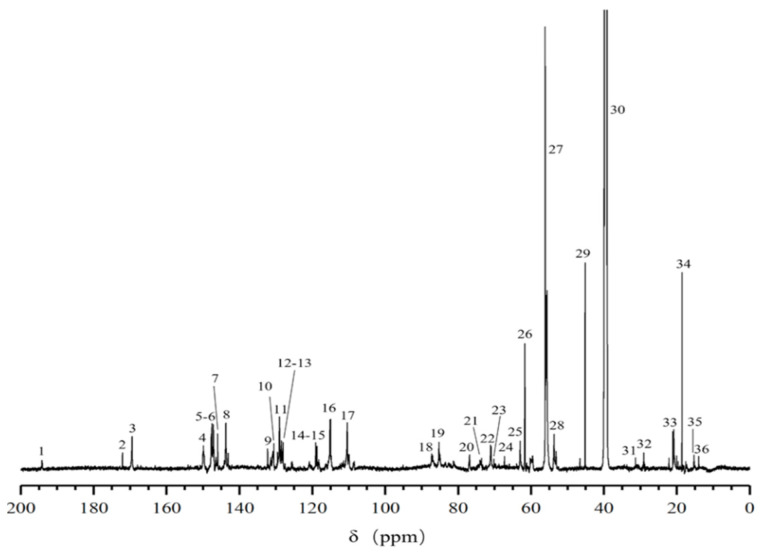
^13^C-NMR spectrum of G-DHP.

**Figure 2 molecules-28-03589-f002:**
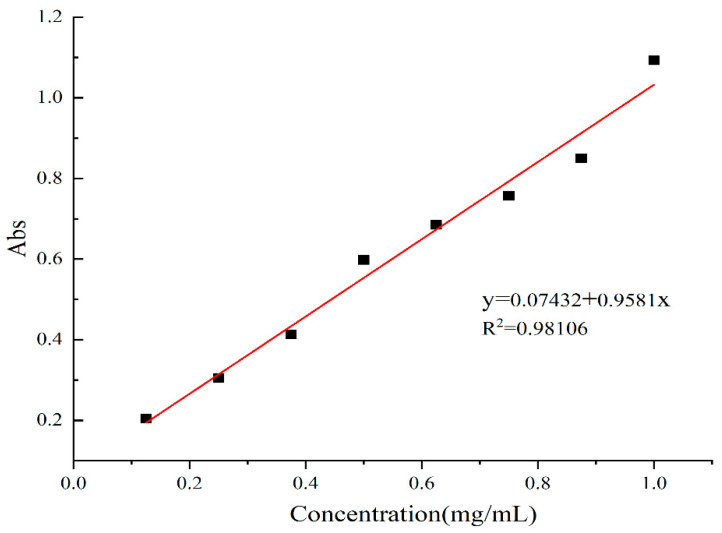
Relationship between gallic acid concentration and absorbance at 760 nm.

**Figure 3 molecules-28-03589-f003:**
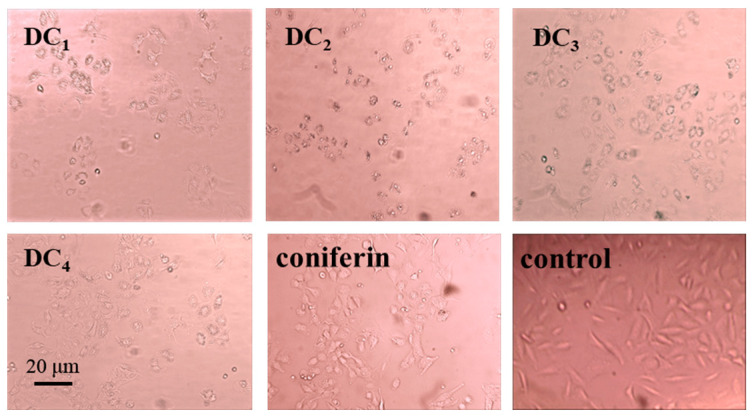
Effects of the classified G-DHP fractions and coniferin on the growth of A549 lung cancer cells at a concentration of 800 μg/mL. Legend: DC_1_–DC_4_: compounds classified from the G-DHP.

**Figure 4 molecules-28-03589-f004:**
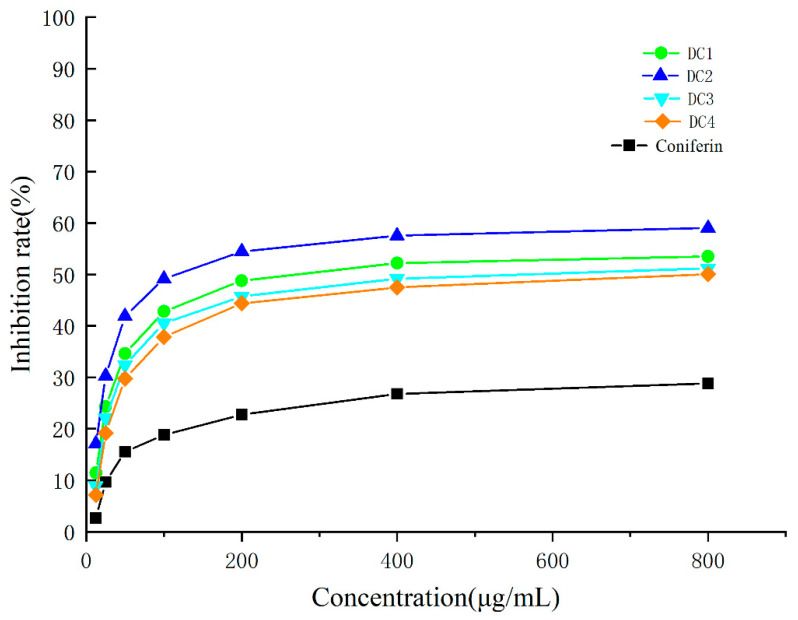
Relationship between different concentrations of classified fractions and coniferin and the inhibition rate of A549 lung cancer cells.

**Figure 5 molecules-28-03589-f005:**
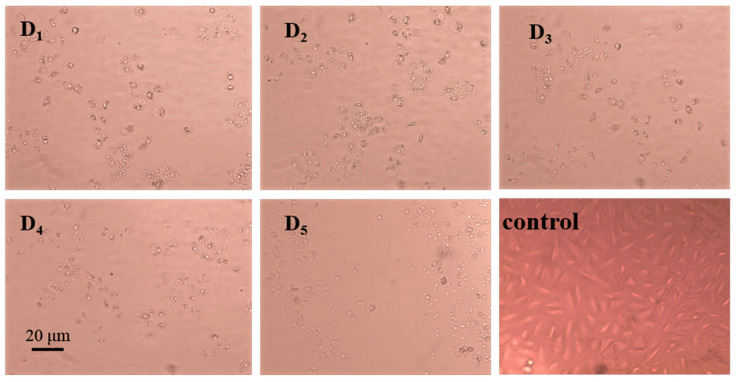
Effects of the purified compounds from the DC_2_ fraction, at a concentration of 800 μg/m, on the growth status of A549 lung cancer cells. Legend: D_1_–D_5_: compounds purified from the ether fraction of G-DHP by preparative chromatography; control: control group without sample and solvent.

**Figure 6 molecules-28-03589-f006:**
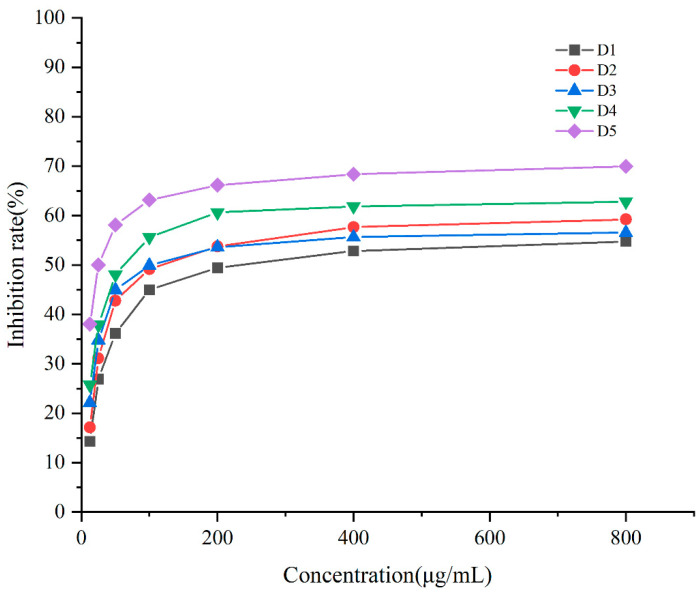
Relationship between different concentrations of purified compounds and the inhibition rate of A549 lung cancer cells.

**Figure 7 molecules-28-03589-f007:**
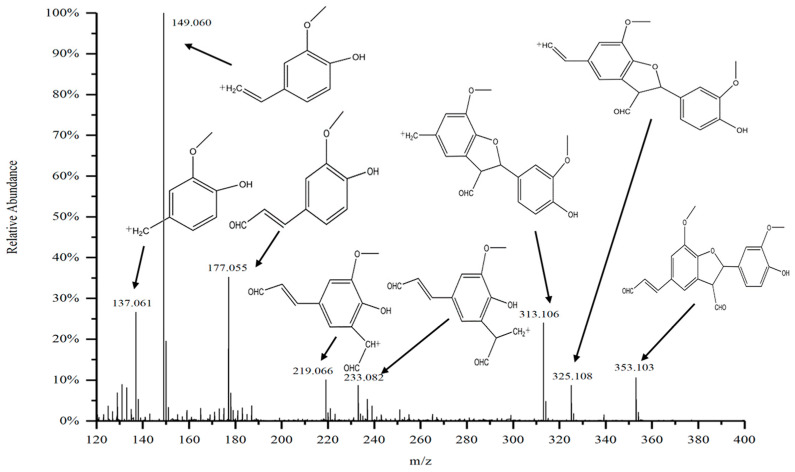
Mass spectrum of purified substance D_5_.

**Figure 8 molecules-28-03589-f008:**
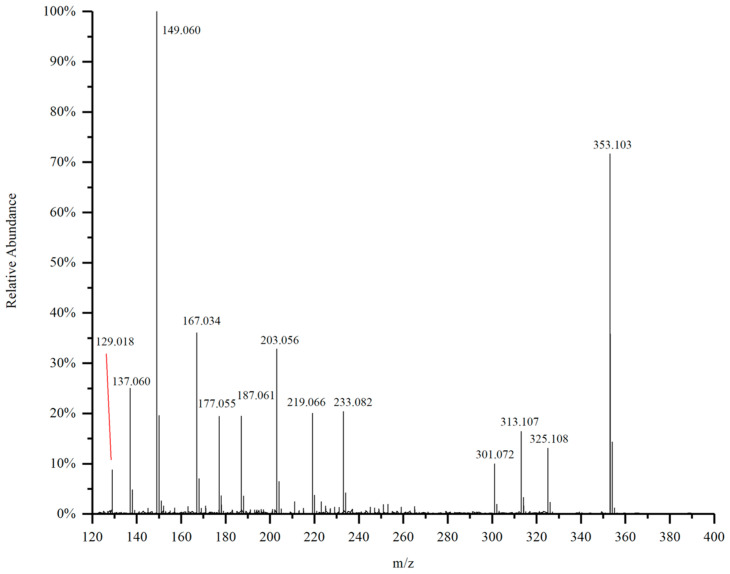
Mass spectrum of purified substance D_4_.

**Figure 9 molecules-28-03589-f009:**
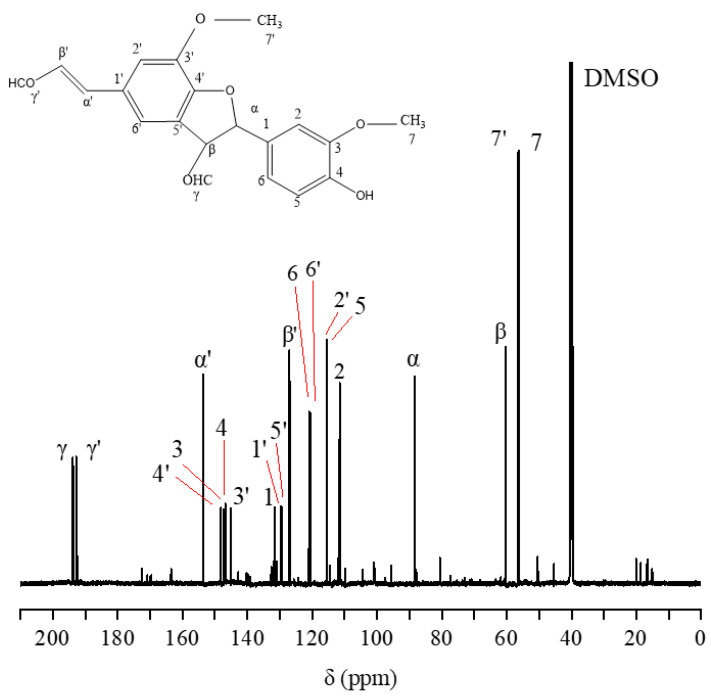
^13^C-NMR spectrum of the purified compound D_5_.

**Figure 10 molecules-28-03589-f010:**
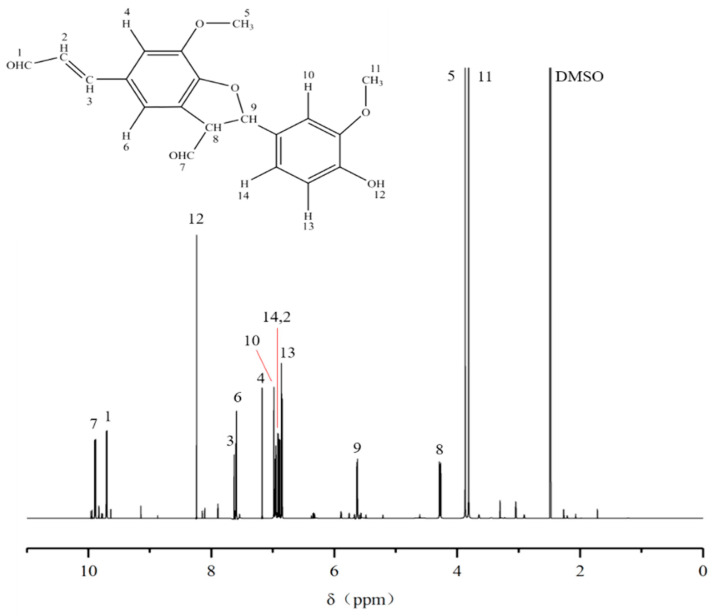
^1^H-NMR spectrum of the purified substance D_5_.

**Figure 11 molecules-28-03589-f011:**
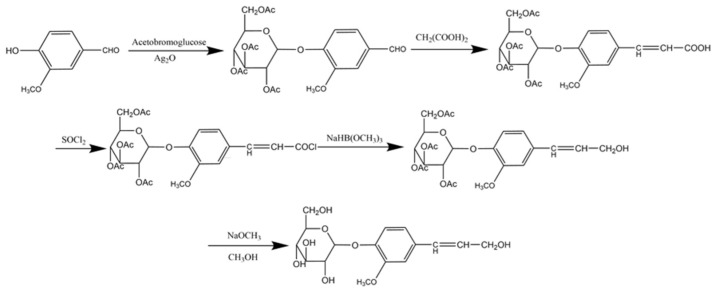
Synthesis route of coniferin.

**Figure 12 molecules-28-03589-f012:**
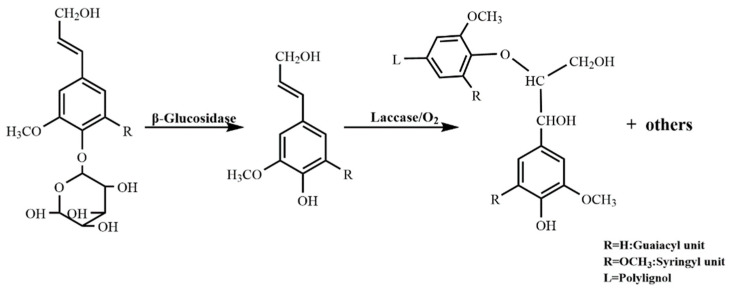
The polymerization of DHP.

**Figure 13 molecules-28-03589-f013:**
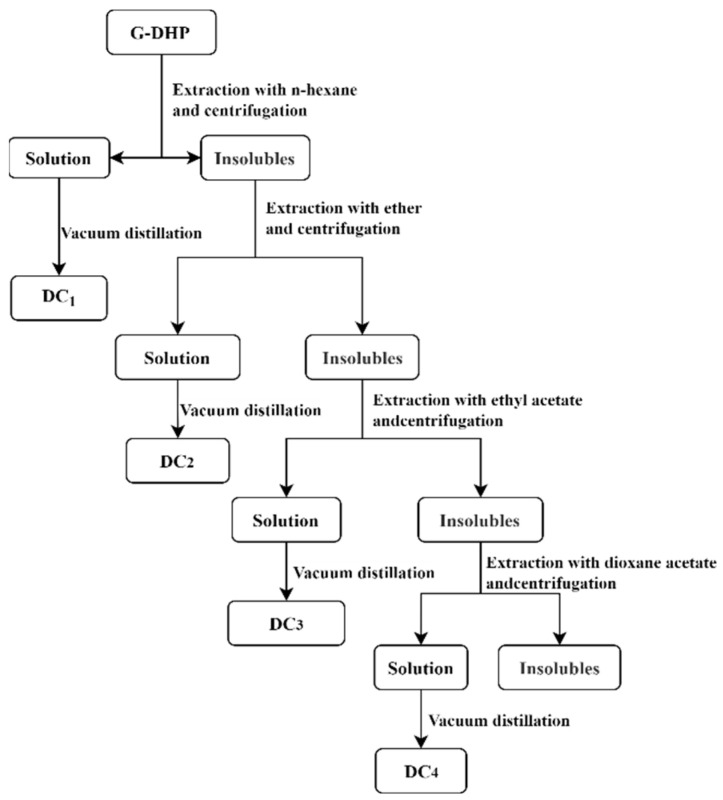
Classification flow chart of the G-DHP.

**Table 1 molecules-28-03589-t001:** Average molecular weights of the DHP fractions.

DHP Fractions	M_w_	M_n_	PDI
DC_1_	188	277	1.47
DC_2_	402	643	1.60
DC_3_	1203	1695	1.41
DC_4_	1853	3486	1.88

Legend: M_w_: weight weighted average molecular weight; M_n_: number average molecular weight; PDI: polymer dispersity index.

**Table 2 molecules-28-03589-t002:** Total phenol contents of the classified fractions of the G-DHP.

DHP Fractions	M_w_	TPC
DC_1_	277	228.66
DC_2_	643	186.57
DC_3_	1695	92.21
DC_4_	3486	70.53

**Table 3 molecules-28-03589-t003:** IC_50_ values of classified G-DHP fractions on A549 lung cancer cells.

Classified Component	IC_50_ (μg/mL)
DC_1_	318.55 ± 53.49
DC_2_	181.46 ± 28.01
DC_3_	401.71 ± 64.27
DC_4_	461.03 ± 79.53

**Table 4 molecules-28-03589-t004:** The IC_50_ values of the purified substances from the DC_2_ fraction on lung cancer A549 cells.

Purified Substances	IC_50_ (μg/mL)
D_1_	286.76 ± 47.16
D_2_	179.25 ± 27.96
D_3_	113.70 ± 23.55
D_4_	61.54 ± 17.10
D_5_	28.61 ± 8.52

## Data Availability

The data presented in this study are available in the manuscript.

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
