# Peer review of "Preparation and Anti-Lung Cancer Activity Analysis of Guaiacyl-Type Dehydrogenation Polymer"

_molecules, 2023, doi:10.3390/molecules28083589_

Round 1

Reviewer 1 Report

Please provide a reference compound for comparison. It is important for readers to know the effectiveness of D3-5. 

Are the activity of D3-5 are significant to inhibit A549 lung cancer cells ?

Any toxicity test e.g. on normal cell ?

Reviewer 2 Report

Zhou et al. described the synthesis of guaiacyl dehydrogenated lignin polymer (G-DHP) using coniferin as a substrate in the presence of β-glucosidase and laccase. The results indicate that the presence of an aldehyde group on the side chain of the phenylpropane unit of G-DHP enhances its anticancer activity. Although the topic is interesting, there are several issues that should be addressed.

  1. The result and presentation of figures 4 and 6 are very poor and not adequate for publication.
  2. A scale bar should be added to figures 3 and 5.
  3. The anticancer activity against normal cancer cell lines should be investigated.
  4. Further anticancer investigation should be addressed, such as the effect on migration and colony formation.
  5. There are many typos and grammar mistakes in the whole manuscript, which should be corrected and proofread.

Round 2

Reviewer 2 Report

The authors have addressed most of my concerns; I recommend acceptance of this paper.